# Sodium Chloroacetate Modified Polyethyleneimine/Trimesic Acid Nanofiltration Membrane to Improve Antifouling Performance

**DOI:** 10.3390/membranes11090705

**Published:** 2021-09-14

**Authors:** Kaifeng Gu, Sichen Pang, Yong Zhou, Congjie Gao

**Affiliations:** Center for Membrane and Water Science & Technology, Zhejiang University of Technology, Hangzhou 310014, China; 2111701057@zjut.edu.cn (K.G.); pangsc1229@126.com (S.P.); gaocj@zjut.edu.cn (C.G.)

**Keywords:** nanofiltration, sodium chloroacetate, modification, antipollution performance

## Abstract

Nanofiltration (NF) is a separation technology with broad application prospects. Membrane fouling is an important bottleneck-restricting technology development. In the past, we prepared a positively charged polyethyleneimine/trimesic acid (PEI/TMA) NF membrane with excellent performance. Inevitably, it also faces poor resistance to protein contamination. Improving the antifouling ability of the PEI/TMA membrane can be achieved by considering the hydrophilicity and chargeability of the membrane surface. In this work, sodium chloroacetate (ClCH_2_COONa) is used as a modifier and is grafted onto the membrane surface. Additionally, 0.5% ClCH_2_COONa and 10 h modification time are the best conditions. Compared with the original membrane (M0, 17.2 L m^−2^ h^−1^), the initial flux of the modified membrane (M0-e, 30 L m^−2^ h^−1^) was effectively increased. After filtering the bovine albumin (BSA) solution, the original membrane flux dropped by 47% and the modified membrane dropped by 6.2%. The modification greatly improved the antipollution performance of the PEI/TMA membrane.

## 1. Introduction

Nanofiltration (NF) membranes have seen huge developments in the past few decades [1,2,3,4]. There are many materials used as NF membranes, including polysulfone (PSF) [5], polyethersulfone (PES) [6], polyamide (PA) [7], polyimide (PI) [8], etc. The membrane structure can be symmetrical or asymmetrical. Commercial NF membranes have been widely produced and applied to some substance purification and separation links. As an advanced separation technology, the treatment effect of NF membranes is excellent, but some aspects still need improvement. The problem of membrane fouling in some complex systems is a concern of many researchers [9,10,11,12]. Recently, many new antipollution NF membrane studies have been reported. Wang et al. prepared an automated PENF membrane surface with a millimeter-size pattern to improve membrane performance [13]. Chen et al. adopted an in situ photo-grafting strategy to graft the bactericidal polyhexamethylene biguanide and hydrophilic polyethylene glycol onto the polyamide membrane surface. The resulting composite membrane showed great antibacterial and antifouling performance [14]. Membrane fouling will reduce the permeate flux and worsen the effluent quality. The cost of process operation will increase substantially as a result, which greatly limits the application of NF membranes. Although some chemical cleaning methods can be used to solve the problem of membrane fouling, it will inevitably increase operating costs and reduce work efficiency. Moreover, the membrane structure may be damaged and reduce the service life. 

There are many reasons for the formation of membrane fouling, such as scale, suspended particles, and organic matter adsorption [15,16,17]. Especially for NF membranes, researching the pollution process is difficult due to its complex separation mechanism. For example, the surface of the NF membrane can be charged, uncharged, positively charged, or negatively charged [18,19,20]. This means that many filter media (protein or other polyelectrolytes) may become the main source of pollution. Many related studies have been carried out to improve the antifouling ability of membranes. Surface modification is a relatively common antifouling strategy for NF membranes [21,22]. Under physical or chemical action, some special materials can be grafted onto the membrane surface. The interface characteristics (hydrophilicity and chargeability, etc.) of the membrane surface are therefore changed. In this way, many studies have successfully improved the antifouling ability of PA, graphene oxide (GO), cellulose, NF membranes, etc. [23,24,25,26,27].

Polyethyleneimine (PEI) has become an ideal material for preparing NF membranes due to its excellent film-forming properties and high chargeability. Usually, the surface of the PEI NF membrane is positively charged and it has a high removal capacity for multivalent cations. In the past, we prepared positively charged PEI/TMA NF membranes with excellent performance [28]. In the separation of salt solutions, the PEI/TMA membrane showed great selectivity and permeability, but in the antipollution experiment of protein, the membrane flux dropped drastically in a short period of time. The hydrophilicity and chargeability of the membrane surface led to this result. In this work, sodium chloroacetate (ClCH_2_COONa) was grafted to the PEI/TMA membrane surface to improve the antifouling ability. The hydrophilicity of the membrane was improved and the positive charge was weakened slightly. The PEI/TMA membrane flux was improved and the BSA adsorption phenomenon was greatly alleviated. 

## 2. Experiments

### 2.1. Materials

These chemical reagents were provided by Aladdin (Shanghai, China), including polyethyleneimine (PEI, Mw = 70,000, 50% solution), trimesic acid (TMA, AR, 98%), sodium chloroacetate (ClCH_2_COONa, AR, ≥98%), sodium laurylsulfonate (SDS, AR, 98%), and bovine serum albumin (BSA, Biotech, 96%). Magnesium chloride (MgCl_2_, powder, ≥99%), sodium chloride (NaCl, AR, 99.5%), magnesium sulphate (MgSO_4_, AR, ≥98%), sodium sulphate (Na_2_SO_4_, AR, ≥99%), and anhydrous ethanol (AR, ≥99.7%) were purchased from China National Pharmaceutical Group Corporation. NaOH (≥96%) and HCl (AR, 36%~38%), purchased from Sigma-Aldrich (Shanghai, China), were used to adjust the pH of the feed liquid. Polyethersulfone ultrafiltration membrane (PES, MWCO 30,000~40,000 Da), provided by Huzhou Research Institute, was used as the base material of the composite membrane. 

### 2.2. Membrane Preparation and Modification

The PEI/TMA NF membrane was prepared by a gradient cross-linking process [28,29]. The TMA cross-linking agent solution infiltrated the PES membrane surface, which can cross-link the PEI solution to form a separation layer. The preparation details can be found in previous research [28]. Here, the membrane, prepared with parameters of 1.0% PEI, 0.1% TMA, cross-linking temperature of 90 °C, and a heating time of 10 min, was marked as M0. The membrane M0 was immersed in the ClCH_2_COONa aqueous solution. The modification process is shown in Figure 1. The M0 membranes immersed with 0.1%, 0.2%, 0.3%, 0.4%, and 0.5% ClCH_2_COONa aqueous solutions for 2 h were marked as M0-1, M0-2, M0-3, M0-4, and M0-5. The M0 membranes immersed with 0.5% ClCH_2_COONa aqueous solution for 2 h, 4 h, 6 h, 8 h, and 10 h were marked as M0-a, M0-b, M0-c, M0-d, M0-e. ClCH_2_COONa was easily ionized under the action of static electricity and the ClCH_2_COO^-^ could be successfully grafted to the surface of the membrane.

### 2.3. Characterization

The membrane surface morphology could be observed by field emission scanning electron microscopy (FE-SEM, Hitachi Limited, SU8010, Tokyo, Japan). Before observation, the membrane surface was sputtered with nano-platinum gold by an ion sputtering instrument. Atomic force microscopy (AFM, BRUKER, Dimension Icon, Los Angeles, CA, America) was used to analyze the surface roughness of the membrane. In tapping mode, the cantilever scanned above the sample in air. Due to the interaction between the sample and the cantilever, the cantilever swung in the vertical direction, and the surface morphology of the sample was reflected. The hydrophilicity of the membrane surface was measured by a contact angle tester (OCA50AF, Data physics, Filderstadt, Germany). Before the test, the membrane was smoothly pasted on the glass slide with double-sided tape. An amount of 1 μL of deionized water touched the membrane surface lightly, and the dynamic changes of the droplets were recorded by the camera. The membrane surface potential was obtained by a solid surface zeta potential analyzer (SurPASS, Anton Paar GmbH, Graz, Austria). The pH of the test system was adjusted by 0.1 mol/L NaOH and HCl solutions. A conductivity meter (DDSJ-308A, Shanghai, China) was used to evaluate the concentration of a single salt solution.

### 2.4. Membrane Performance Evaluation

The evaluation of membrane performance mainly focused on selectivity and permeability. The membrane was encapsulated in a cell with a 0.00196 m^2^ effective filtration area and tested by cross-flow filtration. The membrane was pre-pressed at 2 bar for 1 h before testing, and then membrane performance was evaluated at 2 bar. The test system was controlled at room temperature. The permeate flux was calculated by the formula F=V/(A × Δt), V (L) was the permeate volume, A (m^2^) is the effective filtration area, and ∆t is the filtration time. The solute rejection was calculated by the formula ${R=(1-C}_{p}{/C}_{f})\times 100\%$, C_p_ is the permeate concentration, and C_f_ is the feed liquid concentration. The concentration of a single salt solution could be replaced by conductivity. The flux drop rate (F_r_) was defined by flux after filtering for 80 min (F_p_) and initial flux (F_0_). It was calculated by the formula $F_{r}=\left(1\;-\;\frac{F_{p}}{F_{0}}\right)\times 100\%$.

## 3. Results and discussion

### 3.1. Membrane Surface Morphology Analysis

The original membrane (M0) and the modified membranes’ (M0-1–M0-5) SEM images were observed at 10 kV voltage. The surface morphology of the membrane had hardly changed (Figure 2), which was due to the fact that ClCH_2_COONa is a small molecule substance and the membrane pores might not have changed (Appendix A) during the modification process. The basic structure of the membrane had not changed. It could be considered that the composite membrane was still intact after modification. In addition, the modification could not change the surface roughness (Figure 3, Table 1). The undulating valley ridge structure encouraged some organic molecules to accumulate in the valley [30,31], and this accumulation was difficult to wash away. A smooth membrane surface was conducive to slow the formation of the cake layer. The modification failed to achieve results in this respect, which means that protein adsorption still existed.

### 3.2. Membrane Interface Characteristics

Although the physical morphology of the surface had not changed, the hydrophilicity could be significantly affected. In the static contact angle analysis, with the extension of the ClCH_2_COONa concentration and modification time, the water contact angle decreased significantly (Figure 4). The wettability of water molecules on the membrane surface was improved. It could be believed that the highly hydrophilic membrane surface could improve the antifouling performance [32,33]. The hydrophilic area made it easy to form a dense hydration layer on the membrane surface, which weakened the adsorption of pollutants during the filtration process. 

The reduced contact angle indicated that the ClCH_2_COONa was successfully assembled on the surface of the membrane. The introduction of hydrophilic carboxyl groups led to this result. In addition to hydrophilicity, chargeability was also affected. The zeta potentials of the original PEI/TMA membrane and the modified membrane were analyzed (Figure 5). The original PEI/TMA membrane (M0) surface was highly positively charged, which was due to the highly protonated amino group. Sodium chloroacetate was easily ionized in aqueous solution to form negatively charged CH2COO^-^ groups, while the amino groups on the PEI/TMA membrane surface were protonated and positively charged. After the PEI/TMA membrane was immersed in the sodium chloroacetate solution, by extending the immersing time (Appendix A), the CH2COO^-^ groups were gradually attracted to the membrane surface via electrostatic interaction. In addition, influenced by negatively charged carboxyl groups, the positively charged amino groups were effectively shielded, and the positive chargeability of the membrane surface was effectively reduced.

### 3.3. Membrane Desalination and Permeability Performance

In the membrane performance test experiment (Figure 6), the PEI/TMA membrane exhibited a stable desalination performance. The rejection rate of the other three salts (NaCl, MgSO_4_, Na_2_SO_4_) had also not changed much (Appendix A). ClCH_2_COONa could not damage the membrane structure. The SEM images of the membrane surface also confirmed this (Figure 2). As the degree of modification deepened (high ClCH_2_COONa concentration and extension of modification time), water flux rose. In addition, the modification failed to change the molecular weight cut-off (MWCO) of the PEI/TMA membrane (Appendix A). It could be believed that the increased flux was mainly caused by the hydrophilic surface and the neutralization of chargeability [34,35]. In the modification process, the increase in water flux was positively correlated with hydrophilicity (Figure 4). High wettability promoted rapid penetration of water molecules through the membrane. It is believed that the highly hydrophilic membrane surface has a strong antifouling ability [36,37]. This result showed that the modification process not only maintained the composite membrane structure, but the basic permeability of the membrane was also improved.

### 3.4. Membrane Antifouling Test

With the BSA solution (100 ppm, pH 6.0) as the pollution source, the antifouling experiment of the PEI/TMA membrane was carried out (Figure 7). In a short period of time, the PEI/TMA membrane flux dropped sharply, which was caused by BSA adsorption. The membrane surface was blocked and the transport of water molecules was hindered. In addition, during this period of time (20 min), the membrane surface adsorption reached saturation, and the pore plugging phenomenon did not deepen further. The flux did not continue to change too much (Appendix A). After ClCH_2_COONa modification, the BSA pollution had been reduced and the initial flux of the membrane was improved. Increasing the ClCH_2_COONa concentration and extending the modification time could achieve better results. A high concentration of ClCH_2_COONa shielded as many amino sites as possible, which also led to a reduced zeta potential on the membrane surface. It was an effective way to reduce BSA adsorption from the charge effect.

The improved hydrophilicity (Figure 4) promoted faster transportation of water molecules [38,39]. The membrane surface was more likely to form a hydration layer, which hindered the adsorption of BSA molecules. From the SEM image of the membrane surface after filtering the BSA, the adsorption phenomenon on the modified membrane surface was greatly reduced (Appendix A). The reduction in the chargeability of the membrane surface was also a factor in the antipollution of the PEI/TMA membrane [40,41]. Due to the low isoelectric point of BSA (about 4.7), the high positive charge of the original membrane (M0) made the adsorption of BSA easy. After modification, a part of the protonated amino group was shielded [42,43,44] and the charge attraction was weakened between BSA and the membrane surface. Therefore, the blockage of the membrane pores was relieved. The drop rate of the flux had also been reduced (Figure 8). Hence, the modified membrane (M0-1~5, M0-a~e) showed higher water permeability and excellent antipollution ability. The best performance could be achieved after modification with 0.5% sodium chloroacetate for 10 h, and the greater degree of modification hardly worked anymore (Appendix A).

To further verify the pollution resistance of the membrane, the membrane, after filtering BSA, was washed with pure water and the water permeability was tested. After cleaning, the membrane flux recovered to a certain extent (Figure 9), but the degree of membrane flux recovery was not great. It was believed that the adsorbed BSA molecule on the membrane surface was difficult to wash. In addition to the dirt that was physically embedded in the membrane layer, the combination of BSA and the positively charged membrane surface was also powerful. Pure water cleaning could only play a minor role in these pollutions. Through the modification, the initial flux of the membrane and the flux after being polluted had both increased, but membrane fouling still existed. ClCH_2_COONa modification could only reduce but not completely eliminate BSA pollution. Hence, the flux could not be fully recovered.

The acid and alkali environments affect the chargeability of the PEI/TMA membrane surface [45,46], and thus the degree of contamination of the membrane surface by the protein. For the pH 3 BSA solution, the modified membrane still showed strong antipollution ability and high flux (Figure 10a,b). After the modified membrane was contaminated by BSA, the flux drop rate was effectively reduced (Figure 11a,b), and the lowest drop rate was only 6.2%. In an acidic environment, the BSA molecule was positively charged and it was not easily adsorbed on the membrane surface due to charge repulsion [47,48]. Compared with the conventional test environment, the antipollution ability of the modified membrane was improved more obviously. On the contrary, ClCH_2_COONa modification slightly improved the antifouling performance of the PEI/TMA membrane in an alkaline environment (Figure 11c,d). Although the modification increased the initial permeability of the membrane, BSA still reduced the flux in a relatively short period of time (Figure 10c,d). Compared to a lower PH environment, the modification weakened the pollution process to a lesser degree. It was possible that pH 9 was too close to the isoelectric point of the PEI/TMA membrane. Although the modification slightly reduced the positive charge of the membrane, the charge repulsion might not play much of a role in this test environment. Therefore, the adsorption of BSA had not been weakened too much. The modified membrane showed a better antifouling performance in a neutral or weak acid environment.

### 3.5. Performance Comparison

In recent years, there has been more and more research on the preparation of PEI NF membranes. Some newly developed membranes were summarized in Table 2. Combined with this work, the performance of these membranes was compared and analyzed. The PEI/TMA membrane (M0-e) had unique advantages in the removal of MgCl_2_ while the flux reached a high level. In the treatment of water softening and cleaning water, PEI NF membranes could have a great treatment effect. Due to the susceptibility to pollution, the quality of the influent water in the NF process needed to be treated at the front end. High pollution-resistant NF membranes could reduce the cost of the water inlet front. The modified PEI/TMA NF membranes have such characteristics of resistance to BSA protein contamination, therefore it could have very practical application prospects in water softening and household water purification.

## 4. Conclusions

In this work, the PEI/TMA membrane was modified by ClCH_2_COONa through charge action. The modified membrane had higher water flux and pollution resistance. The modification concentration and time were discussed in detail. The membrane structure was not damaged and the initial desalination performance was maintained. Due to the improvement of hydrophilicity and chargeability, the membrane flux had been improved and the degree of BSA contamination had been effectively alleviated. The initial flux of the PEI/TMA membrane increased from 17.2 L m^−2^ h^−1^ to 30 L m^−2^ h^−1^. After filtering the BSA solution, the flux drop rate of the modified membrane under the optimal conditions (0.5% ClCH_2_COONa, 10 h) was only 6.2%. Through this surface modification, the PEI/TMA membrane could achieve a great antipollution effect. It will be beneficial to promote the application of the PEI/TMA membrane in related fields such as protein concentration.

## Figures and Tables

**Figure 1 membranes-11-00705-f001:**
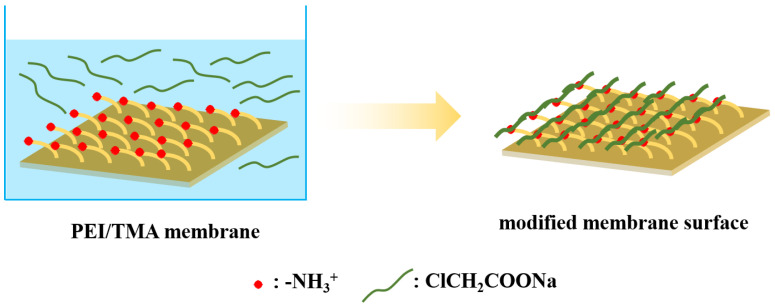
The process of ClCH_2_COONa modifying a PEI/TMA membrane.

**Figure 2 membranes-11-00705-f002:**
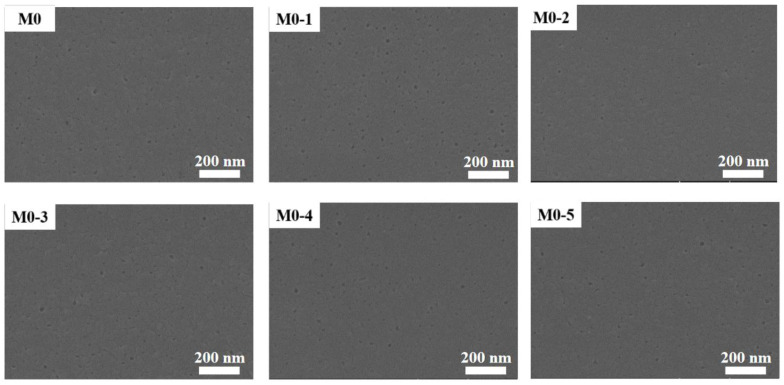
Comparison of membrane surface morphology before and after ClCH_2_COONa modification.

**Figure 3 membranes-11-00705-f003:**
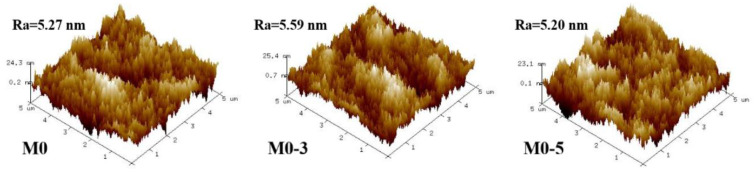
Original membrane and surface roughness (height unit: nm).

**Figure 4 membranes-11-00705-f004:**
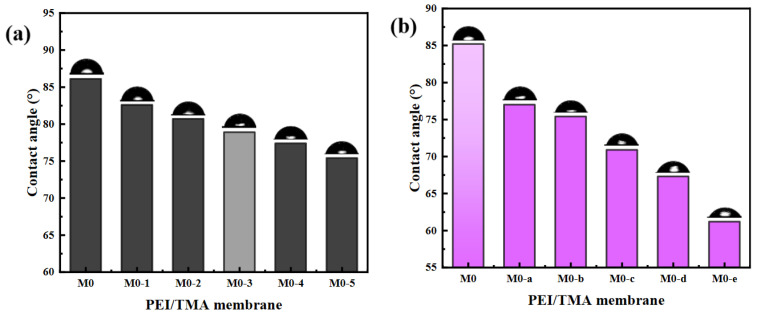
The influence of ClCH_2_COONa concentration (**a**) and modification time (**b**) on the hydrophilicity of the membrane surface.

**Figure 5 membranes-11-00705-f005:**
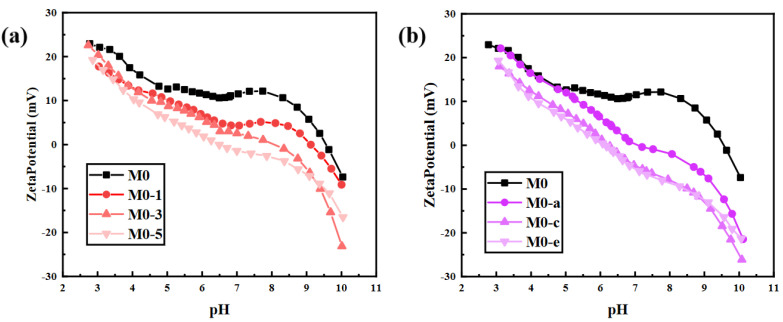
The influence of ClCH_2_COONa concentration (**a**) and modification time (**b**) on the zeta potential of the membrane surface.

**Figure 6 membranes-11-00705-f006:**
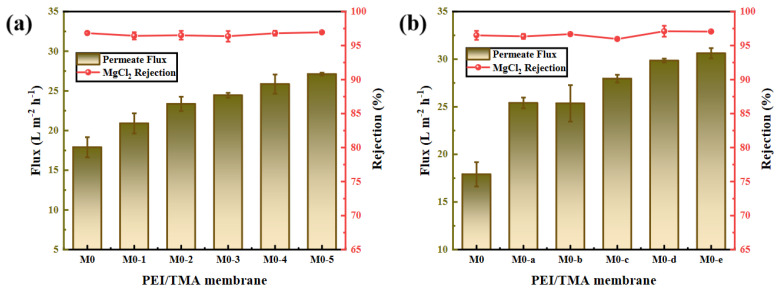
Separation and permeability performance of original membrane and modified membrane ((**a**), ClCH_2_COONa concentration; (**b**), modification time) to 500 ppm MgCl_2_ solution.

**Figure 7 membranes-11-00705-f007:**
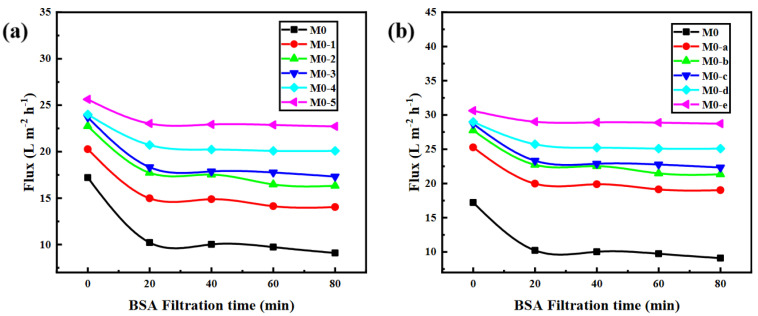
Antipollution test ((**a**), ClCH_2_COONa concentration influence; (**b**), modification time influence) with BSA solution (pH 6.0).

**Figure 8 membranes-11-00705-f008:**
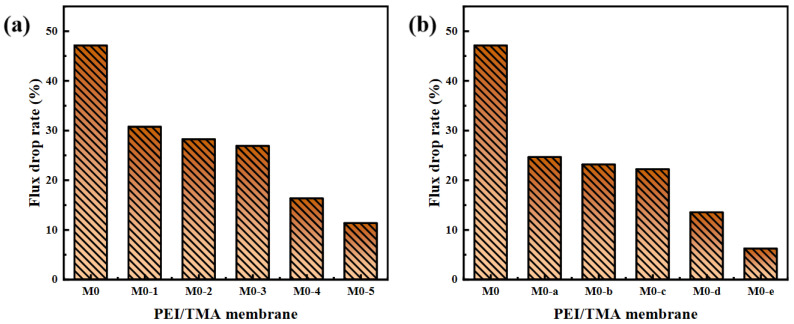
Flux drop rate after membrane fouling ((**a**), ClCH_2_COONa concentration influence; (**b**), modification time influence).

**Figure 9 membranes-11-00705-f009:**
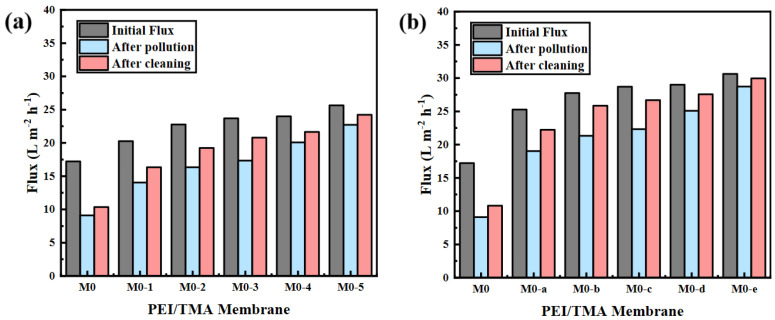
The change of membrane flux ((**a**), ClCH_2_COONa concentration influence; (**b**) modification time influence) after 8 h of pure water cycle cleaning.

**Figure 10 membranes-11-00705-f010:**
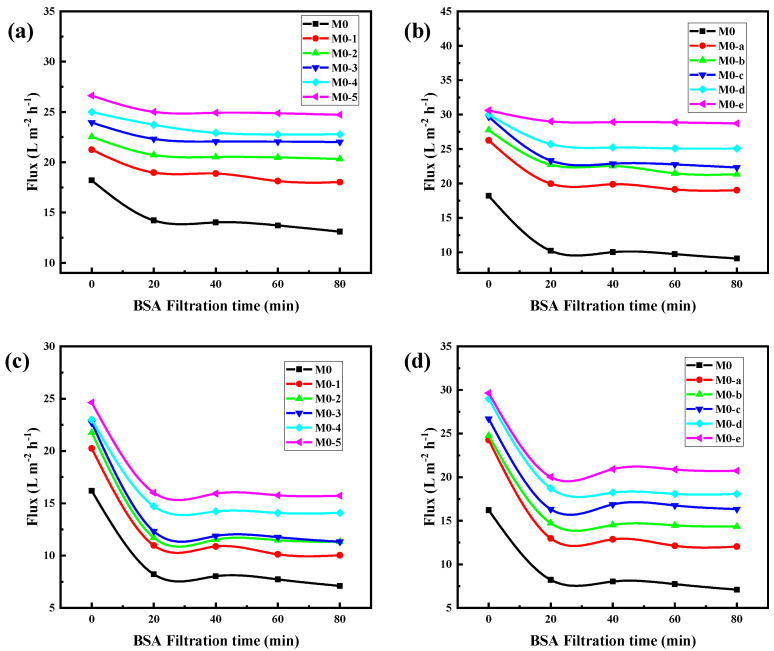
Antipollution test of membranes in acid ((**a**,**b**), 100 ppm BSA solution, pH 3) and alkali environments ((**c**,**d**), 100 ppm BSA, pH 9).

**Figure 11 membranes-11-00705-f011:**
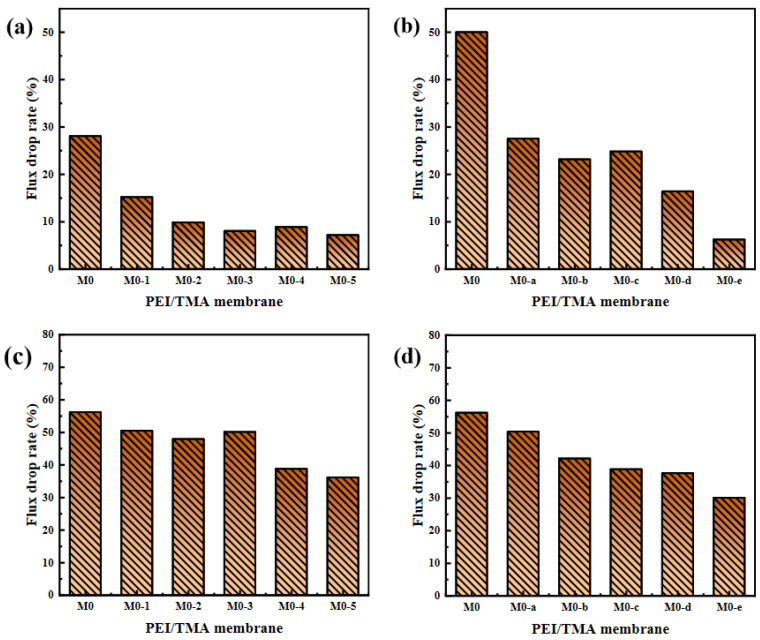
Flux drop rate of membranes in acid ((**a**,**b**), 100 ppm BSA solution, pH 3) and alkali environments ((**c**,**d**), 100 ppm BSA, pH 9).

**Table 1 membranes-11-00705-t001:** Membrane roughness.

Membrane	R_q_ (nm)	R_a_ (nm)
M0	6.71	5.27
M0-3	7.02	5.29
M0-5	6.52	5.20

**Table 2 membranes-11-00705-t002:** Performance comparison with other PEI NF membranes.

Membrane	Flux (L m^−2^ h^−1^ bar^−1^)	Operating Pressure (bar)	MgCl_2_ Rejection (%)	Reference
FPEI/PES	1.1	6	39.3	[49]
PEI/PDA-PAN	2.4	8	92.4	[50]
PEI-TA/PES	40.6	4	9.6	[51]
PEI/C-PES	10.1	2	90	[52]
PS28–Na05–P085	7.4	5.5	93.3	[53]
PEI/TMA M0-e	15	2	96	This work

## Data Availability

Data are available on request.

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
