# Peer review of "Sodium Chloroacetate Modified Polyethyleneimine/Trimesic Acid Nanofiltration Membrane to Improve Antifouling Performance"

_membranes, 2021, doi:10.3390/membranes11090705_

Round 1
Reviewer 1 Report
The author fabricated NF membranes comsisted of PEI/TMA modified with sodium chloroacetate. The fabricated membranes had the resistance against the flux decrease during the filtration of BSA solution. However, the novelty of this researches is unclear compared with the previous studies, especially about the anti-fouling mechanism. And the characterization as NF membranes is not enough. The author should describe the results and discussion in more detail. The detailed comments are as follows.
Whole of the manuscript
The chemistry of the modification with sodium chloroacetate is unclear. Sodium chloroacetate just adsorbed on the PEI/TMA surfaces or chemically reacted with them?
Page 3 Line 12
SDS, MgCl2, NaCl, Na2SO4 and ethanol were not used or described in the manuscripts. What purposes did the author use these reagents?
Page 4 Line 14
Which condition did the author use the AFM apparatus in air or in water? The information about the cantilever is also lacked.
Page 4 final line
How to measure the salt concentration?
Page 5 Line 5.
The author described that the membrane selectivity might not decrease from the SEM observation. However, it is difficult to observe or characterize the nano or sub-nano pores of NF membranes using the SEM or AFM. These descriptions are not appropriate to discuss the selectivity of NF membranes.
Page 7 Fig. 6
The author evaluated only the MgCl2 rejection as the separation performance. Did the modification with sodium chloroacetate affect the selectivity and/or pore size? Was the improvement of the water permeability caused by just only the increased hydrophilicity as described in the manuscript (P7L11)? The rejection against NaCl and Na2SO4, these are frequently evaluated as NF membranes, sould be evaluated to discuss the change of the selectivity and pore size.
How is the separation performance of the fabricated membranes compared with commercial membranes and the NF membranes reported by other reserchers?
Page 8 Fig. 8
How is the stability of the fabricated membranes? Did the author evaluate the separation performance after the filtration tests? Were the anti-fouling properties kept after the filtration tests?
About the fouling states, did the author observe or evaluate the BSA amount adsorbing on the membrane surface?
Page 10 Fig. 10
Was the membrane performance kept during not the BSA filtration test but the long-term preservation at acidic or base conditions?
Author Response
Dear editor, reviewers:
Thank you for giving us the opportunity to submit a revised version of the manuscript. We would like to take this opportunity to express our gratitude to the editors and reviewers. Thank you for your valuable time and comments to help us improve the quality of the manuscript. In the revised manuscript, we try our best to respond to the comments made by the reviewers. We adopted the comments of the reviewer and made the following changes to the contents. To make it easier to find the content, the document pages were marked with line numbers and page numbers. Our respond manuscript and revised manuscript are marked in red.
Reviewer 1
The author fabricated NF membranes comsisted of PEI/TMA modified with sodium chloroacetate. The fabricated membranes had the resistance against the flux decrease during the filtration of BSA solution. However, the novelty of this researches is unclear compared with the previous studies, especially about the anti-fouling mechanism. And the characterization as NF membranes is not enough. The author should describe the results and discussion in more detail. The detailed comments are as follows.
Whole of the manuscript
The chemistry of the modification with sodium chloroacetate is unclear. Sodium chloroacetate just adsorbed on the PEI/TMA surfaces or chemically reacted with them?
Reply:
Fig. S6. XPS O element narrow spectrum analysis on the surface of original membrane and modified membrane.
From the narrow spectrum peak analysis of O element, the peak of carboxyl oxygen on the surface of the membrane was obviously increased with the extension of modification time. The increased carboxyl groups only came from sodium chloroacetate. It could be believed that sodium chloroacetate was tightly adsorbed on the surface of the membrane under the action of electrostatic attraction.
The above content had been added to the supplementary information (SI, Section 5) and some marks were made in the manuscript (Section 3.2, line 155).
Page 3 Line 12
SDS, MgCl2, NaCl, Na2SO4 and ethanol were not used or described in the manuscripts. What purposes did the author use these reagents?
Reply:
Fig. S1. NaCl, MgSO4, Na2SO4 rejection rate of modified membranes.
SDS and ethanol were important parts of the PEI/TMA preparation raw material in the cross-linking agent solution. The detailed PEI/TMA membrane preparation process was shown in previous work [24]. The modification work in this article was based on the prepared PEI/TMA membrane. MgCl2, NaCl, Na2SO4, MgSO4 were all salt solutions for evaluating PEI/TMA membrane desalination performance. The PEI/TMA membrane surface was loose and positively charged, so the desalination rate of the membrane was mainly reflected in the MgCl2. The removal rate of the other three salts was low, and the modification had little effect on this. The test results were shown in Fig. S1.
The above content was added to the supplementary information (Section 1) and some marks had been made in the manuscript (Section 3.3, line 164).
Page 4 Line 14
Which condition did the author use the AFM apparatus in air or in water? The information about the cantilever is also lacked.
Reply:
The AFM images were obtained in tapping mode. The test process was in air. The cantilever scanned above the sample. Due to the interaction between the sample and the cantilever, the cantilever swung in the vertical direction, and the surface morphology of the sample was reflected.
The related descriptions had been added to the manuscript (Section 2.3, line 97).
Page 4 final line
How to measure the salt concentration?
Reply:
For dilute solutions, the relationship between concentration and conductivity was approximately linear. When calculating the rejection rate, the concentration of a single salt solution could be replaced by conductivity. DDSJ-308A conductivity meter was used in the experiment.
The relevant instructions had been added to the manuscript (Section 2.3, line 105; Section 2.4, line 115).
Page 5 Line 5.
The author described that the membrane selectivity might not decrease from the SEM observation. However, it is difficult to observe or characterize the nano or sub-nano pores of NF membranes using the SEM or AFM. These descriptions are not appropriate to discuss the selectivity of NF membranes.
Reply:
We sincerely adopted this suggestion and made relevant corrections.
The original manuscript (Section 3.1, line 118): According to this result, the basic structure of the membrane had not changed, meaning that the membrane selectivity might not decrease.
The modified manuscript (Section 3.1, line 129): The basic structure of the membrane had not changed. It could be considered that the composite membrane was still intact after modification.
Page 7 Fig. 6
The author evaluated only the MgCl2 rejection as the separation performance. Did the modification with sodium chloroacetate affect the selectivity and/or pore size? Was the improvement of the water permeability caused by just only the increased hydrophilicity as described in the manuscript (P7L11)? The rejection against NaCl and Na2SO4, these are frequently evaluated as NF membranes, sould be evaluated to discuss the change of the selectivity and pore size.
Reply:
Fig. S1. NaCl, MgSO4, Na2SO4 rejection rate of modified membranes.
Fig. S3. Molecular weight PEG removal rate curves of membrane M0-d.
The desalination mechanism of the PEI/TMA membrane was dominated by the Donnan effect, so the removal rate for MgCl2 was high. Other salts (NaCl, Na2SO4, etc.) could easily permeate the PEI/TMA membrane. These had been discussed in detail in the past [28]. In this work, the modification failed to make a significant change in the rejection rate of these salts (Fig. S1). The change of NF membrane pore size was usually evaluated by the molecular weight cut-off (MWCO) of PEG. The MWCO of modified membrane M0-d was about 1000 Da (Fig. S3). It was similar to the original membrane M0 [28]. It could be believed that the pore size of the PEI/TMA membrane had not changed significantly. The increase in hydrophilicity and the neutralization of surface chargeability could lead to the rapid transfer of water molecules.
The content in the manuscript has also been modified:
The original manuscript (Section 3.3, line 155): This was mainly caused by the hydrophilic surface [30, 31].
The modified manuscript (Section 3.3, line 168): In addition, the modification failed to change the molecular weight cut-off (MWCO) of the PEI/TMA membrane (SI, Fig. S3). It could be believed that the increased flux was mainly caused by the hydrophilic surface and the neutralization of chargeability [34, 35].
How is the separation performance of the fabricated membranes compared with commercial membranes and the NF membranes reported by other reserchers?
Reply:
In the positively charged NF membrane prepared with PEI, the separation and permeability of the modified PEI/TMA membrane had great advantages. Some comparisons were listed in Table 2. The PEI/TMA membrane (M0-e) had unique advantages in the removal of MgCl2 while the flux could reach a high level.
Table 2. Performance comparison with other PEI NF membranes.
Membrane |
Flux (L m-2 h-1 bar-1) |
Operating pressure (bar) |
MgCl2 rejection (%) |
Reference |
|
FPEI/PES |
1.1 |
6 |
39.3 |
45 |
|
PEI/PDA-PAN |
2.4 |
8 |
92.4 |
46 |
|
PEI-TA/PES |
40.6 |
4 |
9.6 |
47 |
|
PEI/C-PES |
10.1 |
2 |
90 |
48 |
|
PS28–Na05–P085 |
7.4 |
5.5 |
93.3 |
49 |
|
PEI/TMA M0-e |
15 |
2 |
96 |
This work |
|
Page 8 Fig. 8
How is the stability of the fabricated membranes? Did the author evaluate the separation performance after the filtration tests? Were the anti-fouling properties kept after the filtration tests?
About the fouling states, did the author observe or evaluate the BSA amount adsorbing on the membrane surface?
Reply:
Fig. S2. Comparison of membrane performance after BSA filtration.
Fig. S4. Anti-pollution test after 80 minutes.
The fabricated membranes were stable. In fact, after 20~40 min pollution, the membrane performance remained almost unchanged. After filtering the BSA, the original membrane and the modified membrane maintained a high rejection rate of MgCl2 (Fig. S2). It could be believed that within 20 to 40 minutes, the pollution had reached the limit. A longer pollution test was made and the membrane flux no longer decreased (Fig. S4). The anti-fouling properties could be kept. This part of the content had been added to the supplementary information and the marks had been made in the manuscript (Section 3.4, line 187).
Fig. S5. Comparison of M0 and M0-e before and after filtering BSA solution.
From the electron microscope image of the membrane surface after filtering the BSA, the adsorption phenomenon on the modified membrane surface was greatly reduced (SI, Fig. S5).
The SEM images had been added to the supplementary information (SI, Fig. S5) and related marks were also made in the manuscript (Section 3.4, line 195).
Page 10 Fig. 10
Was the membrane performance kept during not the BSA filtration test but the long-term preservation at acidic or base conditions?
Reply:
The membrane performance was kept during the BSA filtration test. The BSA solution was adjusted to pH 3 and pH 9.

Reviewer 2 Report
Manuscript Number: membranes-1357568
Title: Sodium chloroacetate modified Polyethyleneimine/ Trimesic acid
nanofiltration membrane to improve antifouling performance
Type: Research article
Recommendation: Major Revision
Comments to authors: The authors have presented an interesting study about modifying PEI/TMA membrane by ClCH2COONa through charge action. Authors have presented robust results with logical explanation. The work can be considered for publication after addressing the following minor comments.
- Does the membrane modification reduce initial pure flux of water? How does it change the separation properties?
- What prevents chain propagation during the grafting of the ClCH2COONa on membrane?
- Does hydrophilic surface fouls less for all foulant? Or it is case specific?
- Does modification for too long or using too high concentration of modifier molecule effect the performance of the membrane? From graphs it seems that all membrane properties keep on improving. So why stop after a certain amount of modification?
- How do you define flux drop rate?
- Why there is no pore plugging after 20 mins of experiment? (Page 8, discussion of Figure 7)
- How many cycles of washing, cleaning and separation this membrane can survive?
- At what pressure these membranes tested?
Specific comment:
- Abstract: ‘The initial flux of the PEI/TMA membrane could be increased from 17.2 Lmh to 30 Lmh.’ – this statement is confusing. Does this mean a fresh membrane before and after modification has difference in initial flux? Did the flux increase after grafting? It is advisable to rewrite this properly.
- Abstract: It is suggested that optimum membrane be replaced by modified membrane.
- Introduction: what is film forming properties of PEI NF membrane?
- Fig 6: Is this the initial water + salt flux? After what interval was the reading taken after start of the filtration process?
Author Response
Dear editor, reviewers:
Thank you for giving us the opportunity to submit a revised version of the manuscript. We would like to take this opportunity to express our gratitude to the editors and reviewers. Thank you for your valuable time and comments to help us improve the quality of the manuscript. In the revised manuscript, we try our best to respond to the comments made by the reviewers. We adopted the comments of the reviewer and made the following changes to the contents. To make it easier to find the content, the document pages were marked with line numbers and page numbers. Our respond manuscript and revised manuscript are marked in red.
Reviewer 2
The authors have presented an interesting study about modifying PEI/TMA membrane by ClCH2COONa through charge action. Authors have presented robust results with logical explanation. The work can be considered for publication after addressing the following minor comments.
- Does the membrane modification reduce initial pure flux of water? How does it change the separation properties?
Reply:
The membrane modification improved water flux and maintained rejection rate (Fig. 6). The membranes M0-1~M0-5 and M0-a~M0-e were all modified membranes, which were modified on the basis of membrane M0. Due to the improved hydrophilicity of the surface of the modified membrane, the transport of water molecules was accelerated. Although the surface chargeability had decreased, it did not have a big impact on the salt rejection rate.
- What prevents chain propagation during the grafting of the ClCH2COONa on membrane?
Reply:
Maybe the meaning expressed in the manuscript was not clear. The sentence “no chain growth reaction occurred during the modification process”, this meant that sodium chloroacetate was a small molecule, and its contribution to the molecular weight of the polymer separation layer was extremely low. Therefore, it had almost no influence on the sieving effect of the membrane pores. In order not to cause misunderstanding, the following changes were made:
The original manuscript (Section 3.1, line 117): which was due to the fact that ClCH2COONa was a small molecule substance and no chain growth reaction occurred during the modification process.
The modified manuscript (Section 3.1, line 128): which was due to the fact that ClCH2COONa was a small molecule substance and the membrane pores might not be changed (SI, Fig. S3) during the modification process.
- Does hydrophilic surface fouls less for all foulant? Or it is case specific?
Reply:
Membrane fouling was related to membrane surface characteristics and pollutants. Here it was case specific. This work was only for BSA pollution. Hydrophilic surface was beneficial to resist BSA pollution [32, 33].
- Does modification for too long or using too high concentration of modifier molecule effect the performance of the membrane? From graphs it seems that all membrane properties keep on improving. So why stop after a certain amount of modification?
Reply:
Fig. S7. BSA anti-fouling test for modified membranes with higher concentration of sodium chloroacetate (a) and longer modification time (b).
The membranes modified with higher concentration of sodium chloroacetate (0.5% M0-5, 0.6% M0-6, 0.7% M0-7) and the membranes modified for longer time (10 h M0-e, 12 h M0-f, 14 h M0-g) were tested for BSA anti-pollution. The result was shown in Fig. S7. The further modification could not improve membrane performance more effectively. It could be believed that membrane M0-e was the best modified membrane. Hence, a higher degree of modification was no longer needed.
The above content had been added to the supplementary information (SI, Section 6) and some marks were made in the manuscript (Section 3.4, line 203).
- How do you define flux drop rate?
Reply:
The flux drop rate (Fr) was defined by flux after filtering for 80 minutes (Fp) and initial flux (F0). It was calculated by formula .
The above content had been added to the manuscript (Section 2.4, line 116).
- Why there is no pore plugging after 20 mins of experiment? (Page 8, discussion of Figure 7)
Reply:
The surface of the membrane was smooth and flat (Fig. 2, Fig. 3), so the adsorption caused by electrostatic action might be dominant. After enough BSA molecules were adsorbed, no further blockage would occur. The degree of BSA contamination on the membrane surface had an upper limit. After that, the membrane flux no longer dropped. From the pollution test, the pollution process had been completed in 20 min.
- How many cycles of washing, cleaning and separation this membrane can survive?
Reply:
Pure water cleaning was weak for flux recovery, especially for the best modified membrane (M0-e). But the overall anti-fouling performance of the membrane was stable. In the cycle pollution and cleaning test for more than 48 h, the flux of membrane M0-e was always maintained at 28~30 L m-2 h-1.
- At what pressure these membranes tested?
Reply:
The pressure was 2 bar. This had been marked in the manuscript (Section 2.4, line 110).
Specific comment:
- Abstract: ‘The initial flux of the PEI/TMA membrane could be increased from 17.2 Lmh to 30 Lmh.’ – this statement is confusing. Does this mean a fresh membrane before and after modification has difference in initial flux? Did the flux increase after grafting? It is advisable to rewrite this properly.
Reply:
Compared with the original membrane, the initial flux of the modified membrane was increased. The membranes M0-1~M0-5 and M0-a~M0-e were all modified membranes, which were modified on the basis of membrane M0. Modification of sodium chloroacetate improves the initial flux and anti-fouling performance of the membrane. Maybe the description in the manuscript was not clear, the following changes were made.
The original manuscript (Abstract, line 16): The initial flux of the PEI/TMA membrane could be increased from 17.2 L m-2 h-1 to 30 L m-2 h-1.
The modified manuscript (Abstract, line 16): Compared with the original membrane (M0, 17.2 L m-2 h-1), the initial flux of the modified membrane (M0-e, 30 L m-2 h-1) was effectively increased.
- Abstract: It is suggested that optimum membrane be replaced by modified membrane.
Reply:
We sincerely adopted this suggestion (Abstract, line 18).
- Introduction: what is film forming properties of PEI NF membrane?
Reply:
Generally, the surface of the PEI NF membrane was positively charged and it had a high removal capacity for multivalent cation. Some clarifications had been added to the manuscript (Introduction, line 56).
- Fig 6: Is this the initial water + salt flux? After what interval was the reading taken after start of the filtration process?
Reply:
In Fig. 6, the permeate flux was the MgCl2 solution permeate flux. After 1 h pre-pressing, the permeate volume was read for 20 min to calculate the flux.

Reviewer 3 Report
This study investigates the membrane fouling reduction in nanofiltration through modifying the surface of PEI/TMA membrane. The writing is clear. Readers in the fields of wastewater treatment and biofilm formation will be interested in this work. It is recommended for major revision.
Some suggestions are listed below:
- The main hypothesis and research approach (also experimental design) in the introduction were not sufficiently discussed. e.g. recent advance of related works in field, potential mechanism
- The scale of SEM image is difficult for visual inspection. The roughness parameters can be summarized in table.
- Potential improvement and further application can be more discussed.
- Two relevant references about biofilm fouling and shear are recommenced: Quantitative characterization and analysis of granule transformations: Role of intermittent gas sparging in a super high-rate anaerobic system; A super high-rate sulfidogenic system for saline sewage treatment
Author Response
Dear editor, reviewers:
Thank you for giving us the opportunity to submit a revised version of the manuscript. We would like to take this opportunity to express our gratitude to the editors and reviewers. Thank you for your valuable time and comments to help us improve the quality of the manuscript. In the revised manuscript, we try our best to respond to the comments made by the reviewers. We adopted the comments of the reviewer and made the following changes to the contents. To make it easier to find the content, the document pages were marked with line numbers and page numbers. Our respond manuscript and revised manuscript are marked in red.
Reviewer 3
This study investigates the membrane fouling reduction in nanofiltration through modifying the surface of PEI/TMA membrane. The writing is clear. Readers in the fields of wastewater treatment and biofilm formation will be interested in this work. It is recommended for major revision.
Some suggestions are listed below:
The main hypothesis and research approach (also experimental design) in the introduction were not sufficiently discussed. e.g. recent advance of related works in field, potential mechanism
Reply:
We sincerely adopted this suggestion and added some discussion.
The original manuscript (Introduction, line 34): For example, the problem of membrane fouling in some complex systems is concerned by many researchers [9, 10].
The modified manuscript (Introduction, line 33): The problem of membrane fouling in some complex systems is concerned by many researchers [9-12]. Recently, many new anti-pollution NF membrane studies had been reported. Wang et al. prepared an automated PENF membrane surface with a millimeter size pattern to improve the membrane performance [13]. A MXene@CA nanofiltration membrane based on Ti3C2TX (MXenes) and cellulose acetate (CA) cross-linking showed improved hydrophilicity, which was conducive to anti-fouling performance [14].
The scale of SEM image is difficult for visual inspection. The roughness parameters can be summarized in table.
Reply:
The SEM images (Fig. 2) had been improved and the roughness parameters had been summarized in Table 1.
Potential improvement and further application can be more discussed.
Reply:
We sincerely adopted this suggestion and added section 3.5 discussion to the manuscript.
Two relevant references about biofilm fouling and shear are recommenced: Quantitative characterization and analysis of granule transformations: Role of intermittent gas sparging in a super high-rate anaerobic system; A super high-rate sulfidogenic system for saline sewage treatment
Reply:
We sincerely adopted this suggestion and added references [11] [12].

Round 2
Reviewer 1 Report
The author revise to the reviewers' comments properly; therefore, this revised paper is acceptable for publication after minor modification.
Introduction, line 33
The author refered the anti-fouling modification using MXenes, but MXenes is a not popular, quite unique material, and have no relevance to this paper. I think this reference is not appropriate.
Section 3.2, line 155
The author added the discussion about the state of the sodium chloroacetate on the modified membrane surface. However, the author should more clearly describe that the sodium chloroacetate adsorbed on the surface via electrostatic interaction.
Author Response
Dear editor, reviewers:
Thank you for giving us another opportunity to revise the manuscript. We tried our best to answer the reviewer’s questions. The following is the detailed response.
The author revise to the reviewers' comments properly; therefore, this revised paper is acceptable for publication after minor modification.
Introduction, line 33
The author refered the anti-fouling modification using MXenes, but MXenes is a not popular, quite unique material, and have no relevance to this paper. I think this reference is not appropriate.
Reply:
We sincerely adopted this suggestion and replaced the reference 14. The following changes were made:
The original manuscript (Introduction, line 37): A MXene@CA nanofiltration membrane based on Ti3C2TX (MXenes) and cellulose acetate (CA) cross-linking showed improved hydrophilicity, which was conducive to anti-fouling performance [14].
The modified manuscript (Introduction, line 37): Chen et al. adopted an in-situ photo-grafting strategy to graft the bactericidal polyhexamethylene biguanide and hydrophilic polyethylene glycol onto the polyamide membrane surface. The resulting composite membrane showed great antibacterial and anti-fouling performance [14].
Section 3.2, line 155
The author added the discussion about the state of the sodium chloroacetate on the modified membrane surface. However, the author should more clearly describe that the sodium chloroacetate adsorbed on the surface via electrostatic interaction.
Reply:
We sincerely adopted this suggestion and expanded the relevant content.
The original manuscript (Section 3.2, line 155): Under the action of static electricity, sodium chloroacetate was tightly bound to the surface of the membrane and XPS narrow-spectrum analysis proved this (SI, Fig. S6). By the influence of negatively charged carboxyl groups, the positively charged amino groups were effectively shielded, and the positive chargeability of the membrane surface was effectively reduced.
The modified manuscript (Section 3.2, line 155): Sodium chloroacetate was easily ionized in aqueous solution to form negatively charged CH2COO- groups, while the amino group on the PEI/TMA membrane surface was protonated and positively charged. After the PEI/TMA membrane was immersed in the sodium chloroacetate solution, by extending the immersing time (SI, Fig. S6), the CH2COO- groups were gradually attracted to the membrane surface via electrostatic interaction. Besides, by the influence of negatively charged carboxyl groups, the positively charged amino groups were effectively shielded, and the positive chargeability of the membrane surface was effectively reduced.

Reviewer 2 Report
The authors have addressed my comments properly and the manuscript can be published now.
Author Response
Thanks for your comments and suggestions.
Reviewer 3 Report
Accepted
Author Response

(The authors gave the same response as above.)
